# In Silico Analyses of the Role of Codon Usage at the Hemagglutinin Cleavage Site in Highly Pathogenic Avian Influenza Genesis

**DOI:** 10.3390/v14071352

**Published:** 2022-06-21

**Authors:** Mathis Funk, Anja C. M. de Bruin, Monique I. Spronken, Alexander P. Gultyaev, Mathilde Richard

**Affiliations:** 1Department of Viroscience, Erasmus Medical Center, 3000 CA Rotterdam, The Netherlands; m.funk@erasmusmc.nl (M.F.); a.c.m.debruin@erasmusmc.nl (A.C.M.d.B.); m.spronken@erasmusmc.nl (M.I.S.); a.goultiaev@erasmusmc.nl (A.P.G.); 2Group Imaging and Bioinformatics, Leiden Institute of Advanced Computer Science (LIACS), Leiden University, 2300 RA Leiden, The Netherlands

**Keywords:** highly pathogenic avian influenza viruses, low-pathogenic avian influenza viruses, multi-basic cleavage site, evolution, RNA-dependent RNA polymerase, insertions, stuttering

## Abstract

A vast diversity of 16 influenza hemagglutinin (HA) subtypes are found in birds. Interestingly, viruses from only two subtypes, H5 and H7, have so far evolved into highly pathogenic avian influenza viruses (HPAIVs) following insertions or substitutions at the HA cleavage site by the viral polymerase. The mechanisms underlying this striking subtype specificity are still unknown. Here, we compiled a comprehensive dataset of 20,488 avian influenza virus HA sequences to investigate differences in nucleotide and amino acid usage at the HA cleavage site between subtypes and how these might impact the genesis of HPAIVs by polymerase stuttering and realignment. We found that sequences of the H5 and H7 subtypes stand out by their high purine content at the HA cleavage site. In addition, fewer substitutions were necessary in H5 and H7 HAs than in HAs from other subtypes to acquire an insertion-prone HA cleavage site sequence, as defined based on in vitro and in vivo data from the literature. Codon usage was more favorable for HPAIV genesis in sequences of viruses isolated from species or geographical regions in which HPAIV genesis is more frequently observed in nature. The results of the present analyses suggest that the subtype restriction of HPAIV genesis to H5 and H7 influenza viruses might be due to the particular codon usage at the HA cleavage site in these subtypes.

## 1. Introduction

Birds of the orders Anseriformes (e.g., swans, ducks, geese) and Charadriiformes (e.g., gulls)—referred here to as wild aquatic birds—are the original hosts of influenza A viruses, harboring a vast diversity of sixteen and nine different subtypes of the hemagglutinin (HA) and neuraminidase (NA) surface glycoproteins (H1-16; N1-N9) [1]. From these reservoir hosts, influenza A viruses can spill over to poultry species (e.g., chicken, turkey, quail), in some cases leading to sustained circulation [2]. Infections with most avian influenza viruses (AIVs), defined as low pathogenic avian influenza viruses (LPAIVs), are mostly asymptomatic in wild aquatic birds and mild in spillover poultry [2,3]. However, AIVs from the H5 and H7 subtypes can evolve into highly pathogenic avian influenza viruses (HPAIVs) [2]. Virtually all documented HPAIVs have emerged in poultry, and in these species, they cause severe systemic disease leading to mortality rates as high as 100% [4]. The main virulence determinant of HPAIVs in poultry is the HA proteolytic cleavage site, which is changed from a monobasic cleavage site in LPAIV HAs (arginine (R)/lysine (K)) into a multi-basic cleavage site (MBCS, typically (R_n_/K_n_-)R-R/K-R/K-R) through substitutions or, more commonly, via insertion of additional nucleotides by the influenza virus RNA-dependent RNA polymerase (RdRp) [4]. Cleavage of HA into HA1 and HA2 at this site is essential for the fusogenic activity of HA [5]. HAs with an MBCS can be cleaved by ubiquitous furin-like proteases rather than tissue-restricted LPAIV-activating trypsin-like proteases, supporting the systemic spread of HPAIVs in poultry. Since 1959, there have been 51 independent recorded emergences of AIVs with an MBCS, all of them of the H5 and H7 subtypes [4], with the exception of an H4 virus which remained trypsin-dependent [6].

Although the genetic basis for the conversion from LPAIV to HPAIV is known, i.e., the acquisition of an MBCS in HA, HPAIV genesis is unpredictable because the mechanisms underlying this genetic change remain unknown. It is still unclear why the HA cleavage site region presents with repeated insertion events, nor is it known why MBCS acquisition is almost exclusive to two of the 16 HA subtypes. It has been shown previously that artificial insertion of an MBCS into HA is tolerated for all subtypes but H12, H13, and H16, which have not yet been investigated [7,8,9,10,11]. These artificial MBCS introductions render the HA trypsin-independent but do not always confer an HPAIV phenotype, as that also depends on other parts of the viral genome. This indicates that there are no general structural constraints in the HA protein explaining the absence of MBCS motifs in non-H5/H7 AIVs and that the restriction of HPAIV to the H5 and H7 subtypes might therefore be due to differences and/or constraints at the RNA level.

In some H5 HPAIVs, the MBCS is acquired via nucleotide substitutions, leading to the replacement of non-basic amino acids by basic ones. More commonly, MBCS acquisition results from an insertion of at least six nucleotides coding for two basic amino acids [4]. Substitutions can be explained by the low fidelity of the influenza RdRp [12,13], but the exact mechanisms underlying insertions at the HA cleavage site remain to be elucidated. The analysis of MBCS sequences from naturally occurring HPAIVs led to the formulation of several hypotheses. Firstly, it has been hypothesized that stuttering or backtracking and realignment on homopolymer-rich or realignment-prone sequences results in repeated insertions of a single nucleotide or duplications, respectively [14,15,16,17]. Substitutions, stuttering, and backtracking are not mutually exclusive and can happen in a stepwise manner. In fact, insertions are often accompanied by substitutions, although it is impossible to retrospectively determine the order of events. Secondly, a subset of MBCS acquisitions, so far only documented in H7, are the result of non-homologous recombination (NHR) leading to the insertion of 9 to 30 nucleotides of other viral or host RNAs at the HA cleavage site [18,19,20,21,22]. We focused here on MBCS acquisition via stuttering or backtracking, which has occurred in both H5 and H7 LPAIVs [4].

We and others have suggested that insertions via stuttering/backtracking might be due to secondary RNA structures surrounding the site of insertion in H5 and H7 HAs [14,15,17,23,24,25,26,27,28]. Indeed, HAs from these subtypes present putative conserved stem-loop structures in which the loop encompasses the site of insertion [23,26,29]. It has been hypothesized that the RdRp might be slowed down or immobilized by RNA structures acting either as a roadblock [14,15] or closing around the RdRp [26], leading to stuttering or backtracking and insertions in the loop sequence. However, HAs of LPAIVs from other subtypes that have so far never evolved to HPAIVs present similar conserved structures (e.g., H6) [26], suggesting that the presence of a stem-loop is not sufficient for insertion generation at the HA cleavage site. It has also become clear that the nucleotide sequence of the region coding for the HA cleavage site has a strong impact on the generation of insertions, with sequences rich in adenines (A; cRNA orientation) and particularly with long stretches of As being insertion-prone [24,28,30,31,32,33,34,35,36]. Serial passaging experiments have revealed that insertions are more easily acquired in H5 and H7 viruses containing substitutions, leading to more A-rich R or K codons at the HA cleavage site than viruses having a consensus LPAIV motif [28,30,31,32,33,34,35]. It has been shown using in vitro minigenome assays that increasing the number of As in the loop region of H5 and H9 HAs via substitutions (and thereby the number of basic amino acids) significantly increases the likelihood of insertion introduction by the RdRp [24,28,36].

Based on these data, we hypothesized that H5 and H7 LPAIV HA sequences have cleavage site sequences that are inherently more insertion-prone than those of other subtypes, explaining the subtype restriction of HPAIV genesis. A previous study by Nao et al. found that HA cleavage site sequences of H5 and, to a lesser degree, H7 HAs tended to contain more purines than those of H4, H6, H9, and H16 [24], but this analysis was only performed on a subset of subtypes and sequences. Here, we established and analyzed a comprehensive dataset of all available full-length LPAIV HA sequences of all 16 subtypes to ascertain if cleavage site sequences of H5 and H7 HAs could be considered more insertion-prone than those from other HA subtypes, and whether these subtypes can be set apart from the others on a sequence level. Additionally, since substitutions seem to have occurred in most MBCS acquisitions, it was crucial to consider not only the nucleotide composition of HA cleavage site sequences, but also how easily nucleotides could be substituted to generate A-rich insertion-prone sequences. We therefore also analyzed the proximity to insertion-prone sequences via substitutions of all available LPAIV HA cleavage site sequences. We found that H5 and H7 cleavage site sequences stood out by their high number of adenines or purines and that they required less substitutions to reach insertion-prone motifs than cleavage sites of other subtypes, suggesting that the type restriction of HPAIV genesis is linked to the particular codon usage at the LPAIV cleavage site.

## 2. Materials and Methods

### 2.1. HA Sequence Dataset Creation and Curation

All available avian HA sequences marked as full-length were downloaded from the GISAID database (https://www.gisaid.org/, [37]) and the NIAID Influenza Research Database (IRD, https://www.fludb.org/, [38]) in February 2021. Sequences were initially sorted by HA subtype as indicated in the databases and processed using Aliview [39] and Python with the Biopython [40] package as follows. First, open reading frames (ORFs) were extracted and ORFs without stop codon or smaller than 90% of the subtype average were discarded as incomplete. To reduce sampling biases and remove duplicates of sequences present in both databases, identical ORFs from the same or consecutive years (according to the year in the isolate name) were grouped together and a single representative was kept. Sequences with cleavage sites bearing insertions or tetrabasic motifs in P4 to P1 (P1/P1′ corresponding to the R*glycine (G) HA cleavage site, the asterisk denoting the cleavage) were sorted out. After curation, 20,448 LPAIV sequences remained in the dataset.

To account for errors in the subtype classification in the online databases, all 20,448 ORF nucleotide sequences were aligned using the super5 algorithm of MUSCLE v5 [41] and an approximately maximum-likelihood phylogenetic tree was constructed using FastTree2 [42] using the -gtr and -gamma parameters. The tree was then annotated using iTOL (https://itol.embl.de/, accessed on 24 January 2022) [43], and sequences with subtypes not matching their position in the tree were re-annotated accordingly. After re-annotation, a new tree was built using the same parameters to verify that all sequences clustered in the right subtype (Figure 1).

HA cleavage site sequences (position P14 to P8′) were extracted from each ORF using a script looking for conserved amino acid sequences flanking the cleavage site on each side: the highly conserved GLFGAIAG motif in P1′ to P8′, the P in P6 or P7, and the LA/MA in P14 to P13 or P15 to P14 (Appendix A). Sequences in which these motifs were not found were analyzed by hand.

The continent of isolation of each sequence was derived from metadata available from GISAID, and the IRD and virus sequences were sorted into two regions: the Americas for sequences isolated in North and South America, and the Africa–Eurasia–Oceania (AEO) region for all others. These regions correspond mostly to separately evolving AIV geographical lineages which are defined by migratory bird flyways, though there is limited mixing between regions [1]. The species of isolation was derived from the isolate name, and sequences were sorted into four categories: terrestrial poultry, Anseriformes/Charadriiformes, other, and not determined (Appendix A).

### 2.2. HA Cleavage Site Sequence Logos

Sequence logos representing the frequency of each nucleotide or amino acid in the P14 to P8′ or P6 to P1′ region of the HA cleavage site for each subtype were generated with Python using the Biopython [40], LogoMaker [44], and Matplolib [45] packages. Purine/pyrimidine logos were generated after replacing A/G or T/C by R or Y, respectively, in all sequences.

### 2.3. Adenine or Purine Count and Maximum Adenine or Purine Stretch Length Analyses

The nucleotide count and maximum stretch length analyses were performed using Python with the Biopython [40] and pandas [46] packages. The total number of adenines (A) or purines (A or G) in the P4 to P1 region was determined individually for each sequence of each subtype. The length of the longest A or purine stretch was determined by finding all stretches of two or more using the regular expression syntax (A + A)* for A stretches or ([AG] + [AG])* for purine stretches and keeping the length of the longest one. Results of less than two were grouped into the <2 category before plotting.

### 2.4. Analysis of the Number of Substitutions Necessary to Obtain a Tribasic Cleavage Site

The analysis of the number of substitutions necessary to obtain a tribasic cleavage site was performed using Python with the Biopython [40] and pandas [46] packages. For each sequence, each codon of the P4 to P1 region was compared to the target basic amino acid codons for that position (P4 to P3: [AAA, AGA, AGG, AAG], P1: [AGA, AAG]) to determine the lowest number of substitutions required to reach a target codon at each position. The minimal number of substitutions required to have three basic amino acids between P4 and P1, including an AGA or AAG R in P1, was then calculated by summing the substitutions required at the P1 position and the two lowest numbers of substitutions required in P4, P3, or P2.

### 2.5. Analysis of the Number of Substitutions Necessary to Obtain the Longest Possible Adenine Stretch

Analysis of the number of substitutions necessary to obtain the longest possible A stretch was performed using Python with the Biopython [40] and pandas [46] packages. First a comprehensive list of codons found at each position of the P4 to P1 region was established for each subtype by scanning through all sequences of the dataset. For each subtype, every combination of codons in P4 to P1 was then assembled and added to a list of target motifs if it fulfilled two criteria: (a) the P1 codon was an R codon, and (b) the motif contained an A stretch longer than or equal to a set threshold. The threshold was varied from 3 to 10 (10 As is the longest stretch achievable in 4 codons with an R in P1). The P4 to P1 region of each sequence was then compared to each of the target motifs for the corresponding subtype, and the number of substitutions required to reach the target was calculated. For each sequence, the lowest number of substitutions required was then kept and plotted.

### 2.6. Creating the Static and Interactive Graphs for All Analyses

Static graphs in the main text were plotted using the Matplotlib [45] and Seaborn [47] packages and reworked in Adobe Illustrator. These figures represent all sequences of the dataset. The interactive graphs of Appendix A were plotted using Plotly [48], and the HTML file was generated using jinja2 (https://palletsprojects.com/p/jinja, accessed on 7 April 2022). Since the focus of Appendix A was to highlight differences observed between terrestrial poultry and aquatic wild bird sequences, only sequences sorted into these two species categories (which represented the vast majority of the data, Appendix A) were represented. Sequences sorted into the other or not determined categories were discarded. The code used to plot both static and interactive figures (including the jinja template file) is made available at https://github.com/dr-funk/LPAIV-HA-dataset-2022 (accessed on 20 May 2022).

## 3. Results

### 3.1. A Complete Annotated Dataset of 20,448 Full-Length LPAIV HA Sequences across All AIV Subtypes

All available avian HA sequences annotated as full-length were downloaded from GISAID and the Influenza Research Database [37,38]. Duplicate sequences were removed, and ORFs were extracted. In addition, ORFs shorter than 90% of the average length or without a stop codon were discarded as incomplete. To reduce sampling bias, all identical sequences isolated in the same or consecutive years were grouped, and a single representative was kept in the dataset. Sequences were aligned using MUSCLE [41], and phylogenetic HA ORF trees were constructed using FastTree2 [42] to ensure HA sequences were sorted correctly by subtype (Figure 1). Since the mutations leading to HPAIV genesis have always occurred between the P5 and P1′ codons of the cleavage site (P1/P1′ corresponding to the R*G cleavage site) [4], the P6 to P1′ region was extracted from the ORFs, and only cleavage sites containing no insertions were kept to focus on LPAIV sequences. As tetrabasic cleavage sites, i.e., MBCSs acquired via substitution, have been observed in some HPAIV HAs [4], 51/1220 H5 HA sequences with a tetrabasic MBCS were also excluded. No tetrabasic MBCSs were observed in any other subtype than H5. As the analyses performed here are not dependent on RNA orientation, cRNA orientation has been used throughout the manuscript for ease of visualization of the corresponding amino acid sequence.

The majority of the datasets split by HA subtype contained between 1000 and 2000 sequences, with the exception of some outliers (Figure 2, Appendix A). The numbers of available unique sequences for H14 and H15 were very low (33 and 15, respectively), limiting the power of subsequent analyses, while that of H9 was very high (7301). It is important to note that most of the available H9 HA sequences were from viruses isolated in recent H9N2 outbreaks in chickens in China [49]. A remarkable diversity in P5 to P2 motifs was observed between subtypes, whereas sequence conservation was high within each subtype (Figure 2). As expected, the trypsin-like proteolytic cleavage site (R*G/K*G in P1-P1′) was highly conserved within each subtype. The region immediately upstream of the cleavage site also showed strong subtype-specific amino acid conservation, although a few positions were variable in HAs of some subtypes (e.g., P2 and P5 in H7 or P2 and P4 in H12). Besides the R*G/K*G pair in P1 and P1′, there was a strong conservation of a proline (P) in P6. In the H7, H10, and H15 subtypes, this residue was instead present at P7 (Appendix A). Six of sixteen subtypes had two conserved basic amino acids in the P4 to P1 region, most commonly at the P4 and P1 positions themselves. Strikingly, the strong subtype-specific conservation was also present at the RNA level, indicating a very strong codon conservation hinting at the importance of RNA sequences and/or structures in this part of the genome for AIV biology. For H5 in particular, each codon from P5 to P1 was highly conserved, even for R residues which can be encoded by six possible codons. Strong codon conservation across the HA cleavage site region was also observed in H1, H2, H8, and H13 HAs.

For some HA subtypes, the strong codon conservation was specific to the region and/or the species of isolation. To reflect this, the dataset was split by region (the Americas/Africa–Eurasia–Oceania, AEO [1]) and species category (terrestrial poultry, wild aquatic birds, other, not determined, see Appendix A). In general, there were more sequences available from viruses from the Americas than the AEO region (except for H6, H7, H9, H13, and H15) and more sequences of viruses isolated from wild aquatic birds than terrestrial poultry or others (except for H7 and H9), as shown in Appendix A. Splitting the dataset revealed differences in codon usage, for example, between species of isolation in H5 and H10 HAs, and between geographical region of isolation in H7 HAs (Figure 3). H5 cleavage site sequences from viruses isolated from terrestrial poultry tended to have more basic K amino acids in P5 to P1 than those isolated from aquatic wild birds, in accordance with a previous observation by Luczo et al. [50]. H7 HA cleavage site sequences from viruses isolated in the Americas had more cytosine nucleotides in P5 to P1 than those isolated in the AEO. In H10 sequences, the codon usage on positions P5 and P4 was strikingly different between sequences of viruses isolated from terrestrial poultry and wild aquatic birds, though at both positions, the second nucleotide of the codon was always a thymine.

### 3.2. H5 and H7 HAs Contain a High Number of Purines at the HA Cleavage Site

Alignment of all available sequences from newly emerged HPAIVs showed that insertions always occur in the P4 to P1 region [4]. Therefore, all subsequent analyses were performed on this stretch of 12 nucleotides/4 amino acids. As mentioned above, it has been shown that the number of As in the region immediately preceding the HA cleavage site is positively associated with acquisition of insertions [24,28,30,31,32,33,34,35,36]. We therefore set out to compare the total number of As present in the P4 to P1 region of all sequences and their disposition into stretches.

Firstly, the number of As in the four codons preceding the cleavage site was determined for all HA subtypes. Considering all subtypes, most sequences contained five (31.6%) or six (27.2%) As. The H5 and H14 subtypes stood out from the others with 84.4% and 69.7% of sequences having ≥8 As, respectively (Figure 4). Of note, only 33 H14 sequences were available. H3 was the only other subtype for which a significant number of sequences contained ≥8 As, although far fewer than H5 and H14 with 27.0%. H7 sequences contained slightly more As than average, but rarely ≥7 As. When splitting the dataset by species and/or geographical region, two trends became visible: sequences from viruses isolated from terrestrial poultry tended to have more As (observed in H3, H5, and H10 subtypes, Appendix A), as did those from viruses from the AEO region (observed in H3, H6, H7, H9, and H11, Appendix A).

RdRp stuttering and realignment could be favored by the presence of long A stretches rather than separate As spread across the P4 to P1 codons, analogous to what is observed during influenza mRNA polyadenylation, which requires a uracil stretch [51]. Accordingly, it has been shown in minigenome assays that significantly more insertions were observed in H5 or H9 HAs with a 10 or 9 A stretch (respectively) in P4 to P1 than when the stretch was interrupted by one or two nucleotides, even though the total number of As varied only a little [24,28,36].

We therefore analyzed the length of the longest A stretch found in the P4 to P1 region of each HA sequence (Figure 5). Taking all 20,448 HA sequences of the dataset into account, the longest stretches in most HAs, including H5 and H7 HAs, were only two or three As long, as observed in 37.8 and 42.7% of sequences, respectively. H12 was the only subtype with a significant number of sequences presenting stretches of ≥3 As, with 41.1% of sequences having a stretch of four As. Longer A stretches were only observed in a minority of sequences from the H3, H5, H6, H7, H9, and H13 subtypes, with notably 4.1% of H5 sequences having stretches of ≥5 As. Interestingly, these H5 sequences were virtually all from viruses isolated from terrestrial poultry (Appendix A), in accordance with the previous observation that H5 sequences of terrestrial poultry viruses tended to have more As in this region. The longest A stretches in H5 sequences from AEO terrestrial poultry viruses also tended to have one more A than those from terrestrial poultry viruses from the Americas (Appendix A). H7 sequences of viruses isolated from poultry also had, on average, longer A stretches than those isolated from wild aquatic birds (Appendix A). When split by region, both the H6 and H7 subtypes showed a clear trend whereby sequences of viruses isolated in the AEO region had longer A stretches than their American counterparts (Appendix A). This observation was reversed for sequences of the H4 and H12 subtypes, albeit to a lesser extent. Not all the geographical region and/or species trends observed in the previous analysis were reflected here since the total number of As and the length of the longest stretch of As were not necessarily proportional.

Taken together, these data showed that, while H5 sequences had a high number of As compared to other subtypes, the longest A stretch length detected in H5 sequences was average except for a few outliers. These outliers were generally found in sequences of viruses isolated from terrestrial poultry, reflecting slight differences in codon usage in these species with notably more basic K amino acids in P4 to P1 (Figure 3a). On the other hand, H7 sequences performed only slightly above average in both metrics and therefore did not stand out. H7 sequences from viruses isolated in the AEO region tended to have more As and longer A stretches than their counterparts. Strikingly, despite average A-stretch length, high numbers of purine residues and long purine stretches were observed in both H5 and H7 HA sequences. Eleven purines were present in the P4 to P1 region of virtually all H5 HA sequences (Appendix A), with a single pyrimidine (T or C) splitting them into two stretches of seven and four (Appendix A). In H7 HA sequences from the AEO region, a stretch of 10 purines was present in this region (Appendix A), preceded by two pyrimidines in the first and second nucleotide positions of P4. Sequences of H7 viruses from the Americas tended to have only seven purines split into a stretch of four and a stretch of three. H10 and H16 sequences also presented longer than average purine stretches, albeit shorter than those of AEO H7 HA sequences with eight and six, respectively. In addition, high numbers of purines were also observed in H14, H3, and H4 with 10, 9, and 9, respectively, but only with a purine stretch of moderate length, i.e., four at most (Appendix A).

### 3.3. H5 and H7 HAs Require Less Substitutions to Acquire an Insertion-Prone P4 to P1 Motif

Most natural MBCS acquisitions were the result of both substitution and insertion events [4]. It is possible that substitutions precede the insertions and change the LPAIV P4 to P1 HA cleavage region into a more insertion-prone sequence, such as sequences with longer A stretches or containing several basic amino acids (using A-rich codons) [24,28,30,31,32,33,34,35,36]. The number of substitutions required to reach an insertion-prone sequence could vary between subtypes. For example, the relatively low average A stretch length in H5 sequences despite a high number of As is due to the distribution of As into three distinct two- or three-nucleotide stretches (Figure 2). Due to this distribution, one or two nucleotide substitutions can significantly increase the length of the A stretch, as evidenced by the presence of a minority of H5 sequences with longer A stretches (Figure 5). Therefore, we hypothesized that fewer substitutions are required in H5 and H7 HAs than in HAs from other subtypes to obtain an insertion-prone cleavage site. To address this point, we performed additional analyses investigating the proximity to insertion-prone sequences via substitution.

First, we assessed how many single nucleotide substitutions were required in LPAIV cleavage site sequences in order to form a tribasic cleavage site motif that could act as stepping stone for further mutational events. Such a motif was defined as containing at least three basic amino acids in positions P4 to P1, excluding histidine, with an R in P1. The R in P1 was observed in every newly converted HPAIV and is necessary for HA cleavage by ubiquitous proteases, which underlies the highly pathogenic phenotype [4]. Pyrimidine-containing R codons were excluded based on codon usage in HPAIVs: in sequences from newly converted HPAIVs, purine-only R codons represent 54/55 (H5) or 56/59 (H7) R codons [4]. Three basic amino acids were chosen as a minimal requirement, as all H7 HPAIV cleavage sites retain the P that was originally present in the LPAIV P4 position, indicating that a tribasic cleavage site (position P3 to P1) might be sufficient for promoting nucleotide insertions. Furthermore, the P3 glutamic acid (E) and P2 threonine (T) of the LPAIV H5 consensus sequence were each detected in 3/18 newly converted H5 HPAIVs (though it cannot be excluded that the P3 GAA/GAG E codon was restored via stuttering/realignment), again indicating that a tetrabasic MBCS might not be essential for creating an insertion-prone cleavage site and that the three basic amino acids do not necessarily need to be consecutive [4]. In accordance, it has been shown that insertions were detected in a non-consecutive tribasic H5 RKTR cleavage site during in vivo passaging experiments in chickens [31,34]. We made the assumption that RdRp substitution rates are identical in all AIVs and that the data obtained on the genetic instability of di- and tribasic cleavage sites in H5, H7, and H9 HAs [24,28,30,31,32,33,34,35,36] also apply to other HA subtypes.

With these requirements, four subtypes stood out: H3, H5, and H9 with over 90% and H7 with 63.9% of sequences requiring a single substitution to obtain a non-consecutive tribasic cleavage site (Figure 6). Furthermore, 99.6% of H7 sequences required two or fewer substitutions, and when the data were split by region of isolation, it became apparent that virtually all AEO H7 sequences required a single substitution and that those from the Americas required two (Appendix A). This is particularly interesting when considering that almost all American H7 HPAIV conversions have occurred via NHR, while most AEO H7 HPAIV conversions occurred after stuttering/backtracking [4]. An even more drastic impact of the geographical region was observed for the H9 subtype, for which a single substitution was required for almost all AEO sequences to obtain a tribasic cleavage site, while four substitutions were needed for 85% of sequences from American-origin viruses. Finally, one fewer substitution to reach a tribasic cleavage site motif was needed for H11, H12, and H16 sequences of viruses from the Americas than for those from AEO. Differences in proximity to a tribasic cleavage site motif depending on the host species were observed in H5, for which fewer substitutions were necessary on average in sequences of viruses isolated from terrestrial poultry—regardless of geographical origin—than of those isolated from wild aquatic birds (Appendix A).

The proximity to a tribasic cleavage site by substitution might be a contributing factor to the subtype-restricted genesis of HPAIVs from H5 and H7 LPAIVs. However, only one substitution was also required to obtain a tribasic cleavage site for over 90% of H3 and H9 sequences. The fact that H3, H5, H7, and H9 HAs require very few substitutions to reach a non-consecutive tribasic site is in accordance with the presence in the dataset of sequences already presenting such a cleavage site (at 0.4%, 4.5%, 1.7%, and 2.5%, respectively, Figure 6, Appendix A). The only other tribasic cleavage site in the dataset was a KRTR motif found in 2/1697 H4 HA sequences (0.1%, Figure 6, Appendix A). Since H3 and H9 LPAIVs have never given rise to HPAIVs, the presence of a tribasic cleavage site might not be sufficient by itself for insertion events. In some subtypes, such as H5 and H7, the HA cleavage site is situated in the loop of conserved stem-loop RNA structures, which have been suggested to play a role in HPAIV genesis [14,15,17,23,24,25,26,27,28]. Interestingly such stem-loop structures are not conserved in avian H3 HAs [26]. While the stem-loop is conserved in the H9 subtype [26], the tribasic cleavage sites obtained through the lowest number of substitutions are non-consecutive tribasic sites interrupted by a non-basic serine TCT/TCA codon in P3. Similarly, for H3 HA sequences, the tribasic cleavage sites requiring the fewest substitutions have an ACC/ACT T codon in P2. In contrast, tribasic cleavage sites obtained following a single substitution in H5 HA sequences contained no (e.g., REKR) or one (e.g., RKTR, ACA T codon) pyrimidine, while those of AEO H7 HA sequences contained no pyrimidines in P4 to P1. Codons rich in pyrimidines interrupt the purine stretch formed by the three basic amino acids in H3 and H9 cleavage sites, potentially reducing the insertion-prone character. As a consequence, additional substitutions are required to obtain an insertion-prone sequence resembling that of H5 and H7, i.e., mainly composed of purines.

A drawback of the previous analysis is that differences in codon usage observed between HA subtypes or species and region of isolation are not taken into account (Figure 2 and Figure 3), as AAA, AAG, AGA, or AGG codons were allowed in any position between P4 and P2, and AGA or AGG in P1. In many subtypes, these specific codons have never been observed at these positions, which means that there is no evidence that viruses with R and K codons at these positions could emerge.

We therefore performed a second substitution analysis that takes the codon usage in nature of each subtype into account. First, a list of codons observed in each P4 to P1 position in each subtype was established (see Appendix A). All possible P4 to P1 motifs were assembled—still requiring an R codon in P1. Subsequently, the motifs in which the longest A stretch was below a set threshold length (varied from 3 to 10) were discarded. For each sequence, the minimal number of substitutions needed to reach the closest P4 to P1 motif which passed the threshold length was determined. This analysis did not require substitutions to codons coding for basic amino acids (except for the R in P1), though they are favored as they are A-rich, and any stretch over four As requires at least one AAA K codon. This is consistent with the aforementioned observation that some HPAIV MBCS sequences still contain non-basic amino acids, suggesting that the precursor leading to HPAIV genesis can contain at least one non-basic amino acid in the P4 to P1 stretch [4].

H5 was the only subtype for which the theoretical maximum of 10 As could be reached, followed by H7 and H6 with a maximum of eight and six consecutive As, respectively (Figure 7a). The longest A stretch possible through substitution to naturally occurring codons was five As in H2, H3, H9, H11, and H12. The shortest possible A stretches were observed for H8 with only two consecutive As at most, while no A stretches could be formed in P4 to P1 in H15, though only 15 sequences were available for this subtype. We compared the number of substitutions required for each sequence of each subtype to obtain the required A stretch length, as the threshold length, and therefore stringency, increases. Figure 7b shows the result obtained using an A stretch threshold length between four and eight for HAs from the H5, H6, H7, H3, and H9 subtypes. H5, H6, and H7 were included, as the longest A stretches could be formed in these subtypes, and H3 and H9 because they stood out in previous analyses. All results for all thresholds and subtypes are shown in Appendix A. Not only were cleavage sites with >5 consecutive As unable to be formed by substitution in sequences from both the H3 and H9 subtypes when constrained by the codon usage observed in nature, but more than one substitution was also necessary in the majority of the sequences to obtain even a five A-long stretch. In contrast, stretches of six As could be obtained via a single substitution for the vast majority of sequences of the H5 subtype and stretches of up to eight As with only two substitutions. For the majority of H6 HA sequences, only two substitutions were needed to generate stretches of six As, the longest possible A stretch in H6, whereas three substitutions were required for the vast majority of the H7 HA sequences to obtain cleavage site motifs containing stretches of 6–8 As.

The analysis was also performed on the datasets split by species and region of isolation. When split by species of isolation, H5 sequences of viruses isolated from terrestrial poultry or wild aquatic birds performed similarly at low A stretch threshold lengths, but as the threshold length increased, sequences of viruses isolated from terrestrial poultry often required one fewer substitution to reach the target sequence than those from viruses isolated from wild aquatic birds (Appendix A). This was particularly striking when a threshold length of six As was considered, for which a single mutation was required in almost all H5 sequences of viruses from terrestrial poultry, while two were needed in nearly all sequences of viruses isolated from wild aquatic birds. Only codons found in H5 sequences of terrestrial poultry viruses allowed for reaching stretches of ≥8 As, again reflecting the differences in codon and amino acid usage in viruses isolated from terrestrial poultry and wild aquatic birds (Figure 3a). In the H7 subtype, the only difference when considering the species of isolation was observed for the longest possible stretch: about 80% of terrestrial poultry virus sequences required only three substitutions to reach eight As, while 50% of wild aquatic bird virus sequences required four. For H3 and H9, even low stretch lengths of four or five As were possible only for one of the two species categories: wild aquatic birds for H3 and terrestrial poultry for H9.

In this analysis, there were no major difference between H7 HA sequences from the Americas or the AEO region. The only substantial difference was observed when considering the longest possible stretch of eight As, where one fewer substitution was needed for sequences of AEO viruses than for those of American viruses. This was due to the different codon usage in P4: CCA is observed in most sequences of AEO viruses, whereas CCC is observed in most sequences of American viruses.

## 4. Discussion

While the mechanism(s) underlying MBCS-yielding insertions in LPAIVs of the H5 and H7 subtypes are still unknown, it is evident that H5 and H7 are peculiar compared to HAs from other subtypes in regard to HPAIV genesis. Here, we used a comprehensive dataset of LPAIV HA sequences to investigate the subtype restriction of HPAIV genesis. Based on previous data from in vitro and in vivo experiments [24,28,30,31,32,33,34,35,36], we defined insertion-prone sequences as sequences containing a high number of As or purines, long stretches of As or purines, or a high number of A/purine-rich codons coding for basic amino acids K and R. Sequences from the H5 (Americas and AEO region) and H7 (AEO region only) subtypes stood out by their high number or long stretches of purine residues in the P4 to P1 region, respectively, when compared to sequences from other subtypes. H5 cleavage site sequences also contained a comparatively high number of As with almost 85% of cleavage sites having ≥8 As. In analyses considering the proximity of each sequence to an insertion-prone tribasic cleavage site via substitutions, both the H5 and H7 subtypes stood out alongside H3 and H9, though in the latter, non-consecutive tribasic cleavage sites were interrupted by pyrimidine-rich codons rather than purine-rich codons as observed in H5 and H7. When restricting substitutions to only codons observed in nature in each subtype, long A stretches (≥7 As) could only be formed in H5 and H7. Taken together, these results suggest that the particularities of H5 and H7 HA cleavage site sequences compared to that of other subtypes are (i) a high purine residue content and (ii) a proximity via substitution to sequences defined as insertion-prone based on experimental data.

A higher purine content of H5 and H7 HA cleavage site sequences has previously been suggested as reason behind the subtype restriction of HPAIV genesis in a study by Nao et al. [24]. Their analysis was restricted to sequences from viruses isolated from ducks and focused on the loop sequence of predicted cRNA structures of only 29 nucleotides surrounding the HA cleavage site. Of note, these loops were of varying size and position and did not necessarily encompass the P4 to P1 region in all sequences. They found that the predicted loop domains of H5 and, to a lesser extent, H7 sequences tended to have a higher proportion of purines than those of H4, H6, H9, and H16, though uniquely observed loop sequence motifs and not their relative abundance were taken into account. Long purine stretches could be an important factor for influenza RdRp stuttering or realignment. Indeed, during replication, a 10 base pair duplex between template and product is formed in the active site cavity of the influenza RdRp [52]. Following stuttering or realignment, this duplex will contain mismatches unless it occurs in a stretch of repeated nucleotides [53]. However, guanine–uracil base pairs can form in RNA molecules, meaning that A stretches with interspersed guanines (ergo stretches of purines) might be more prone to stuttering/realignment than A stretches interrupted by any other nucleotide, as the uracil stretch of the product RNA has more leeway to realign, as suggested previously by Perdue et al. [15]. This suggested importance of purine stretches might explain why American H7 HPAIVs rarely emerge following stuttering/realignment. AEO H7 HA cleavage sites have a long uninterrupted purine stretch in P3 to P1, while H7 HA sequences from viruses isolated in the Americas have two pyrimidines in P2 (Figure 3 and Appendix A), which would lead to mismatches between the template and product RNA in the RdRp cavity during realignment. Similarly, the P4 to P1 motifs of H3 and H14, which had a high A content (Figure 2 and Figure 4), and the tribasic cleavage site motifs formed in H3 and H9 upon a single substitution were interrupted by at least two pyrimidines in P3 and/or P2 (Figure 6). The disposition of pyrimidines in these sequences might be highly detrimental to insertions via stuttering/realignment.

There was a strong conservation both at the nucleotide and protein level in the HA cleavage site (P4 to P1), but also in the surrounding region (P14 to P8′), within each subtype (Figure 2 and Appendix A). The strong subtype-specific conservation of codons in P14 to P8′ together with the overall conservation of RNA structures [26] hints at a crucial role for RNA sequences and structures in this region and therefore significant evolutionary constraints at both the RNA and the protein level. Perdue et al. suggested that the RdRp might be blocked by RNA structures, leading to stuttering [15]. However, we have shown that the loop of predicted conserved subtype-specific RNA stem-loops encompass the P4 to P1 region and therefore the site of insertion of basic amino acid codons [23,26]. The “road block” model of Perdue et al. cannot explain insertions of nucleotides in the loop, as impeded processivity by the base of the stem-loop would lead to insertions upstream of the cleavage site. We previously proposed an alternative model in which the stem of the stem-loop reforms around the RdRp during replication, leading to the template closing on itself and trapping the RdRp in the loop, causing it to stutter and/or backtrack [26]. Nevertheless, the prediction of putative RNA structures in subtypes other than H5 and H7 [26] suggests that the presence of a stem-loop at the HA cleavage site is not sufficient for insertion events. Results from the present study and others [24,28,30,31,32,33,34,35,36] indicate that an insertion-prone sequence in the loop of the predicted RNA structures is necessary for insertions to occur. The need for both of these factors might explain why HPAIVs have never emerged in subtypes with HAs presenting a conserved RNA stem-loop structure but no insertion-prone sequence at the HA cleavage site (e.g., H2, H6), nor conversely an insertion-prone sequence but poor conservation of the putative RdRp-trapping RNA structure (e.g., H3, H10) [26]. The subtype restriction of HPAIV genesis to H5 and H7 LPAIVs might therefore be primarily due to RNA sequence constraints allowing the formation with only a few substitutions of an insertion-prone sequence contained in the loop of an RNA stem-loop structure. We propose that the bottleneck in HPAIV genesis is the acquisition of substitution(s) that lead(s) to the formation of an insertion-prone loop sequence, which then can be extended by the RdRp via stuttering/backtracking, and that H5 and H7 are unique in that only very few substitutions are required to acquire an insertion-prone loop sequence. This hypothesis is in agreement with in vivo and in vitro passaging experiments showing rapid acquisition of insertions when substitutions leading to a more insertion-prone sequence are introduced in the input HA cleavage site sequence [28,30,32,33,34,35].

Splitting our analyses by region and species of isolation has shed some light on other aspects of HPAIV genesis. Virtually all HPAIV conversion events have occurred in terrestrial poultry species [4]. H5 sequences of viruses isolated from terrestrial poultry contained more As and longer A stretches and required fewer substitutions to reach an insertion-prone sequence than those from viruses isolated from other bird species. The higher A content is also reflected in the amino acid level where sequences from viruses isolated from terrestrial poultry showed a greater frequency of basic K amino acids in P5-P1. A similar observation was made previously by Luczo et al. based on an analysis of H5 HA cleavage sites at the amino acid level [50]. These species-level differences suggest that viruses with tribasic cleavage sites bearing additional basic amino acids (by substitution) might be under positive selection in terrestrial poultry compared to wild aquatic birds. These different amino acid biases in H5 LPAIVs from different hosts lead to RNA sequences with different insertion propensities and might promote the genesis of HPAIV in terrestrial poultry species. No pronounced species-level difference was observed in H7 HA sequences, but clear differences in codon usage in the P4 to P1 region depending on the region of isolation were noticed. As a result, sequences of H7 viruses from the AEO region performed better overall than those from viruses from the Americas in all present analyses. Perhaps these differences are the underlying reason why MBCS acquisitions in American H7 viruses have almost exclusively occurred via NHR rather than stuttering or backtracking [4], though a role for specific RNA structures has also recently been suggested [29].

It is important to note that while the dataset has been curated and annotation errors were corrected by hand where possible, all the analyses of the present study are reliant on the quality of sequences uploaded to the databases and are impacted by biases in avian influenza surveillance and sequencing. For example, African LPAIVs are included in the AEO region, but they are underrepresented and form only a small minority of available sequences. Similarly, for most subtypes, there are far fewer sequences of viruses isolated from terrestrial poultry than from wild aquatic birds, in part due to different levels of circulation in different species. Additionally, this study is mostly focused on requirements at the RNA sequence level rather than at the protein sequence level. Constraints at the protein level, notably on the cleavage efficiency of HA into HA1 and HA2, could not be taken into account here, due to the limited available knowledge for most subtypes. This aspect in particular might have a major impact on the outcome of the substitution analyses in Section 3.3, in which the resulting cleavage sites might not be efficiently cleaved in vivo. In addition, this study does not consider whether minority variants acquiring insertion-prone HA cleavage sequences after substitutions would present a selective advantage over the majority LPAIV population. More research is needed to understand the impact of substitutions in the HA cleavage site at the protein level in regard to cleavage efficiency and virus fitness.

Here, we combined previously obtained insights on RNA structures at the HA cleavage site [23,26] and on the insertion propensity of different P4 to P1 sequences in vitro and in vivo [24,28,30,31,32,33,34,35,36] with sequence data from a comprehensive LPAIV HA dataset. Based on our analyses, we propose that the subtype restriction of HPAIV genesis to H5 and H7 viruses may be due to their particular codon usage resulting in the combination of conserved RNA structures, long purine stretches, and close proximity via substitutions to insertion-prone P4 to P1 sequences. These substitutions might be the major bottleneck in HPAIV genesis, and therefore the fact that more substitutions would be needed in non-H5/H7 HA sequences to acquire an insertion-prone P4 to P1 sequence might restrict HPAIV genesis in these subtypes. In addition, our data suggest that HPAIV genesis might be favored in terrestrial poultry due to host-dependent amino acid biases (possibly resulting from different selective pressures on intermediate tribasic cleavage sites) and that H7 HAs from AEO viruses are more prone to acquire an MBCS following stuttering/backtracking than those from American viruses, as observed in nature. Further studies aiming to understand the minimum sequence and structure requirements resulting in insertions by the RdRp at the HA cleavage site, especially in non-H5/H7 HAs, would help to refine the present analyses and hypotheses.

## Figures and Tables

**Figure 1 viruses-14-01352-f001:**
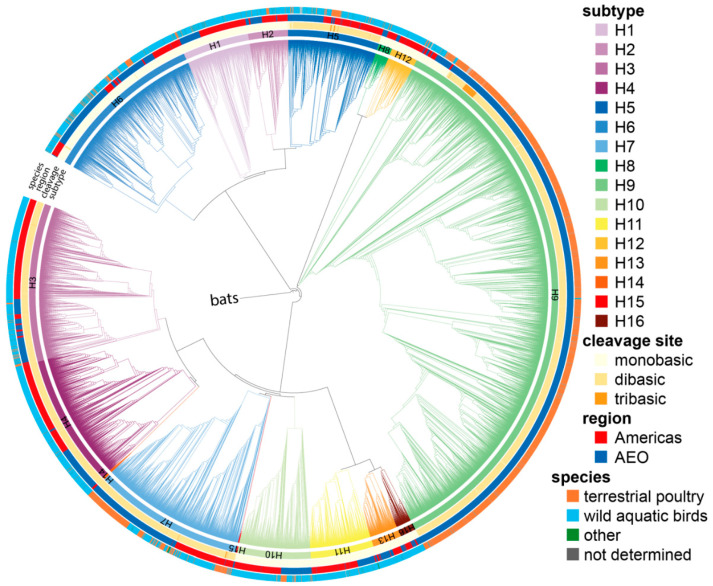
Phylogenetic tree of all 20,448 full-length LPAIV HA nucleotide sequences. The tree was generated after HA subtype reassignment of incorrectly annotated sequences. Branch lengths are arbitrary, and clades are colored by HA subtype using the colors indicated on the right. The clade containing H17 and H18 bat sequences was collapsed and used as root. The inner to outer rings show, respectively: the HA subtype, the number of basic amino acids in the P4 to P1 region (mono-, di-, or tribasic), the region of isolation (Americas or Africa–Eurasia–Oceania, AEO), and the species in which the corresponding virus was isolated (terrestrial poultry, wild aquatic birds, other, and not determined) using the colors indicated on the right.

**Figure 2 viruses-14-01352-f002:**
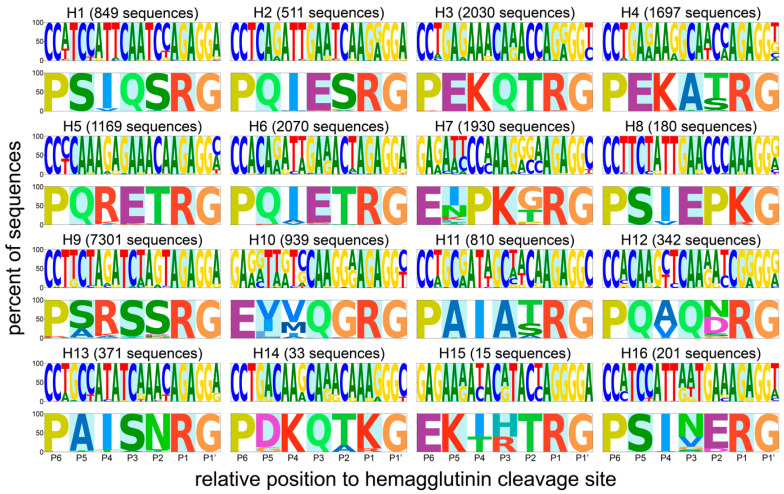
Sequence logos of nucleotide and amino acid composition of HA cleavage site regions (P6 to P1′, P1 position corresponding to the C-terminus of HA1) of different subtypes. For each HA subtype, the number of sequences used to generate the logo is indicated. Letter sizes are directly proportional to the percentage of sequences presenting a given character, as indicated on the y-axes.

**Figure 3 viruses-14-01352-f003:**
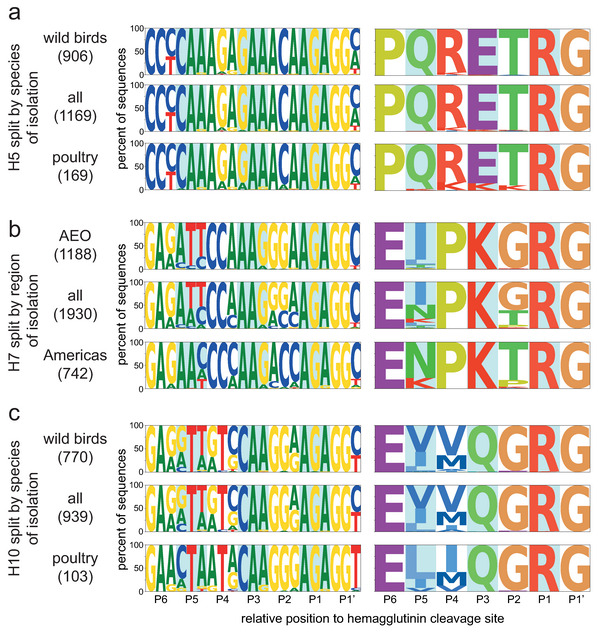
Examples of species- or region-specific codon usage observed in H5, H7, and H10 sequences. (**a**) Sequence logos of sequences from H5 viruses isolated from wild aquatic birds (wild birds), terrestrial poultry (poultry), or all species combined (including 94 sequences from other or not determined species), shown for nucleotide (left) and amino acid (right) sequences. The number of sequences used to generate the logos is indicated in brackets. (**b**) Sequence logos of sequences from H7 viruses isolated in the Americas, the AEO region, or both combined, using the same representation as in (**a**). (**c**) Sequence logos of sequences from H10 viruses isolated from wild aquatic birds (wild birds), terrestrial poultry (poultry), or all species combined (including 66 sequences from other or not determined species), using the same representation as in (**a**).

**Figure 4 viruses-14-01352-f004:**
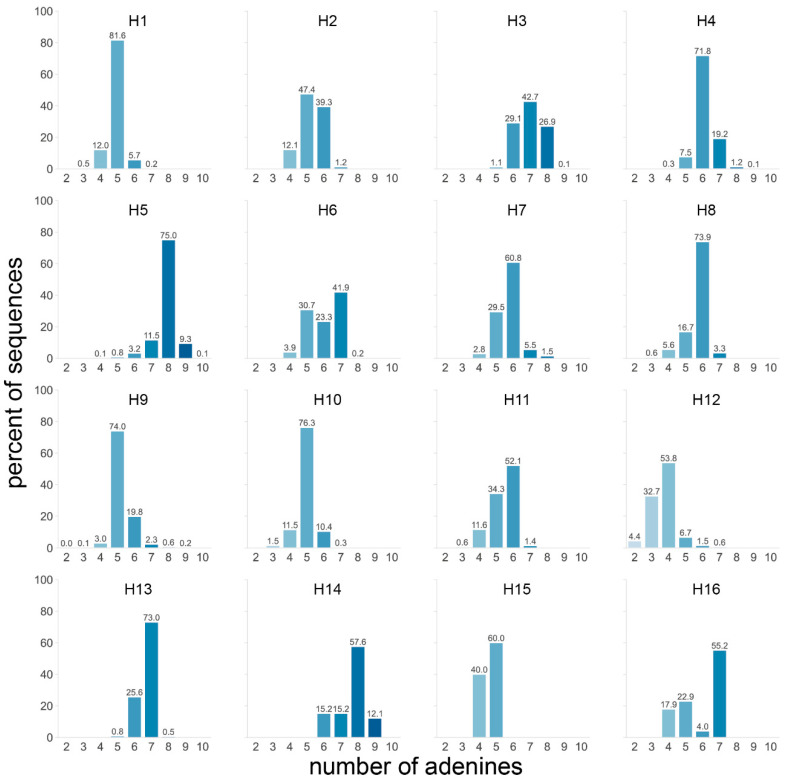
Number of adenines in the P4 to P1 HA cleavage site region for each subtype. The number of adenines in the 4 codons preceding the HA cleavage site was counted in each sequence, and the percentage of sequences with a given number of adenines was calculated independently for each subtype. A darker shade of blue indicates a higher number of adenines, and the exact percentages are indicated above each bar.

**Figure 5 viruses-14-01352-f005:**
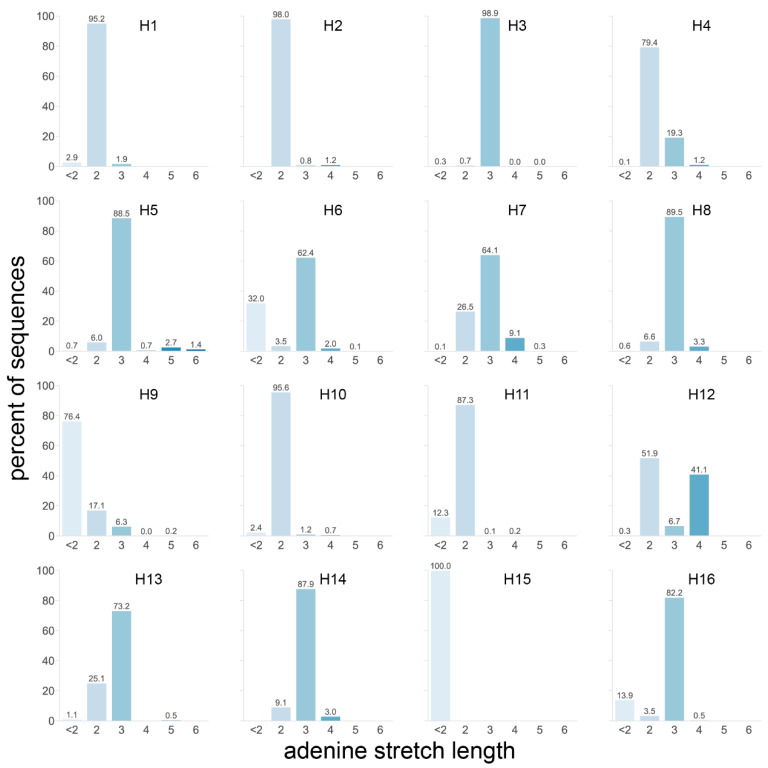
Length of the longest adenine stretch in the P4 to P1 HA cleavage site region for each subtype. The longest stretch of adenines in the 4 codons preceding the HA cleavage site was identified in each sequence, and the percentage of sequences with a given stretch length was calculated independently for each subtype. A darker shade of blue indicates a longer stretch, and the exact percentages are given above each bar.

**Figure 6 viruses-14-01352-f006:**
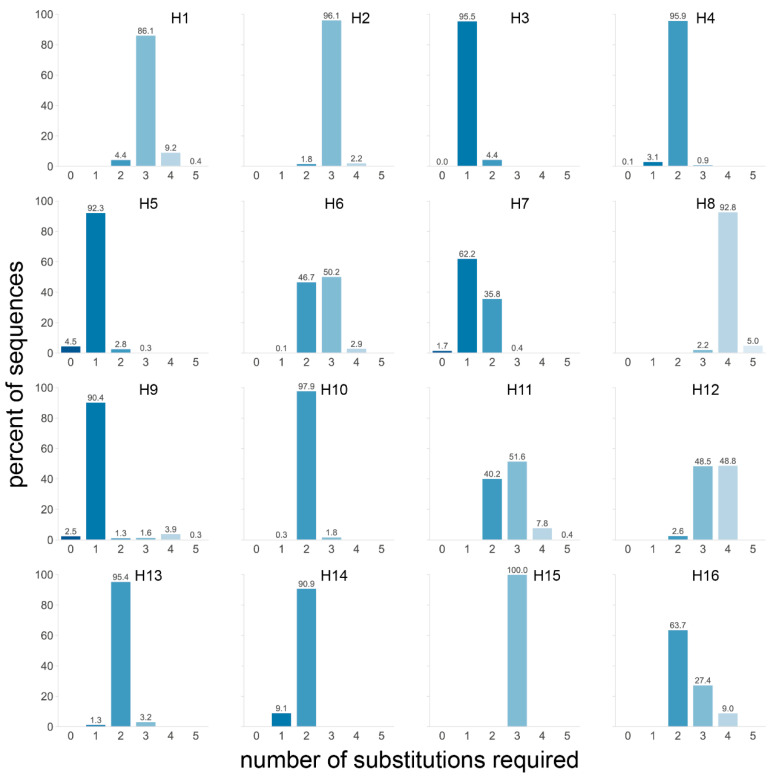
Number of substitutions required in sequences from each subtype to obtain a tribasic cleavage site via substitutions. For each sequence, the number of substitutions required to obtain an arginine in P1 and two other basic amino acids (lysine and arginine only) in P4 to P2 was determined. Only AGG and AGA codons were allowed for arginine, as they represent virtually all arginine codons in HPAIVs in nature. The percentage of sequences requiring a given number of substitutions was calculated independently for each subtype. A darker shade of blue indicates that fewer mutations are required, and the exact percentage is given above each bar.

**Figure 7 viruses-14-01352-f007:**
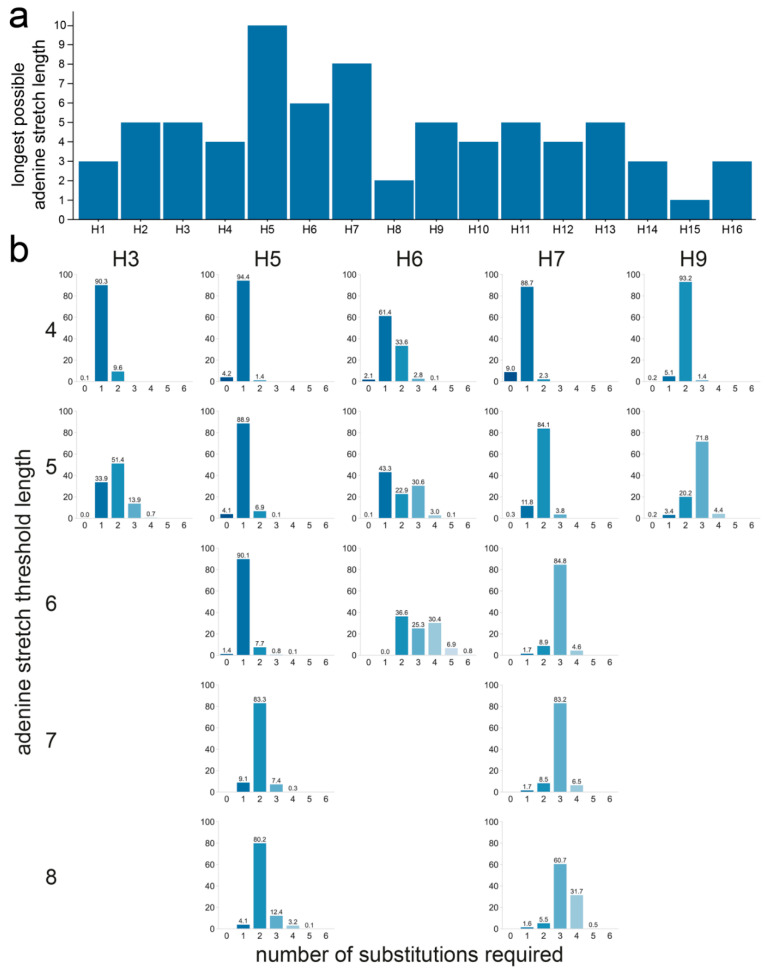
Number of substitutions required to increase adenine stretch length in the P4 to P1 region using only codons found in the dataset at each given position in each subtype. (**a**) Longest adenine stretch achievable with the codon usage of each subtype. (**b**) Number of substitutions required for different adenine stretch lengths for H3, H5, H6, H7, and H9 HAs. For each sequence, the number of substitutions required to obtain a 4- to 8-adenine stretch is shown. The absence of a graph denotes the impossibility of obtaining an adenine stretch of the indicated length using only codons observed in nature. The percentage of sequences needing a given number of substitutions was calculated independently for each subtype. A darker shade of blue indicates that fewer substitutions are required, and the exact percentage is given above each bar.

## Data Availability

All publicly accessible sequences of the dataset are available as Appendix A. For sequences exclusive to GISAID, see Appendix A. Python code for the analyses is made available at https://github.com/dr-funk/LPAIV-HA-dataset-2022 (accessed on 20 May 2022).

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
