# Peer review of "In Silico Analyses of the Role of Codon Usage at the Hemagglutinin Cleavage Site in Highly Pathogenic Avian Influenza Genesis"

_viruses, 2022, doi:10.3390/v14071352_

Round 1
Reviewer 1 Report
This work is complete perfect research that well done and has great importance for science.
Minor notes
Line 254 ….or P1 and P4 in H12). – Probably, it is P2 instead of P1 according to Figure 2?
Line 707 …. Science (80-. ). – Delete (80-. )
Author Response
We thank the reviewers for taking the time to review our manuscript and provide feedback. In response, we have revised the manuscript and provided point-by-point answers be to the reviewer’s comments below.
Reviewer 1
This work is complete perfect research that well done and has great importance for science.
We thank Reviewer 1 for their extremely positive assessment of the manuscript.
Minor notes
Line 254 ….or P1 and P4 in H12). – Probably, it is P2 instead of P1 according to Figure 2?
Line 707 …. Science (80-. ). – Delete (80-. )
We apologize for these mistakes. We corrected them as suggested in the revised manuscript.
Reviewer 2 Report
The submission by Funk et al., entitled ‘In silico analyses of the role of codon usage in H5 and H7 hemagglutinins in highly pathogenic avian influenza genesis’ is a study emphasizing the role of codon usage in highly pathogenic avian influenza genesis. The authors claim that, sequences of the H5 and H7 subtypes stand out by their high purine content 17 at the HA cleavage site. The authors suggest that the subtype restriction of HPAIV genesis to H5 and H7 influenza viruses might be due to the particular codon usage at the HA cleavage site in these subtypes. The manuscript is well written, and well-presented. Even though, the authors supplemented the findings with several bioinformatic analysis, many issues need to be addressed to improve the quality.
Comments
1. Regarding codon usage bias in HPAIV genesis, several studies have been reported. In discussion, the novelty of the current study in comparison to other similar studies should be discussed clearly.
2. Quite common analysis in codon bias study such as Relative synonymous codon usage (RSCU), one of the most widely used parameters for querying the pattern of synonymous codon usage across genes and genomes without confounding influence of the amino acid composition., and effective number of codons (ENC) that estimates the enormity of codon usage bias in a gene were not used. It would have been better if the RSCU and ENC values should be obtained and interpreted.
Author Response
Reviewer 2
The submission by Funk et al., entitled ‘In silico analyses of the role of codon usage in H5 and H7 hemagglutinins in highly pathogenic avian influenza genesis’ is a study emphasizing the role of codon usage in highly pathogenic avian influenza genesis. The authors claim that, sequences of the H5 and H7 subtypes stand out by their high purine content 17 at the HA cleavage site. The authors suggest that the subtype restriction of HPAIV genesis to H5 and H7 influenza viruses might be due to the particular codon usage at the HA cleavage site in these subtypes. The manuscript is well written, and well-presented. Even though, the authors supplemented the findings with several bioinformatic analysis, many issues need to be addressed to improve the quality.
We thank Reviewer 2 for their overall positive feedback.
Comments
- Regarding codon usage bias in HPAIV genesis, several studies have been reported. In discussion, the novelty of the current study in comparison to other similar studies should be discussed clearly.
The present study does not investigate codon bias directly, instead “codon usage” is used to refer to the RNA sequences observed specifically at the hemagglutinin cleavage site in different hemagglutinin subtypes and species (which are directly dictated by different amino acid sequences). We realized that the confusion might have come from the formulation of the title of the manuscript and some turns of phrase in the rest of the manuscript, which might have been misleading in this regard. We have changed the title of the manuscript to “In silico analyses of the role of codon usage at the in H5 and H7 hemagglutinin cleavage sites in highly pathogenic avian influenza genesis” and revised some text in the manuscript (lines 119, 262, 291, 502, 639, and 679) to be more accurate with the used terminology.
Concerning other studies of the codon usage at the hemagglutinin cleavage site, we indeed realized that the study by Luczo et al. (2015) was not referenced. We added it and discussed it in light of our study (lines 383 and 635).
- Quite common analysis in codon bias study such as Relative synonymous codon usage (RSCU), one of the most widely used parameters for querying the pattern of synonymous codon usage across genes and genomes without confounding influence of the amino acid composition., and effective number of codons (ENC) that estimates the enormity of codon usage bias in a gene were not used. It would have been better if the RSCU and ENC values should be obtained and interpreted.
As explained above, codon bias is not directly relevant to our study, especially since the focus of this study is on a very short part of the hemagglutinin gene, the cleavage site. We think that calculating the codon usage bias of a four-codon long region will not add any relevant information for the study, and that the codon usage bias at the level of the whole hemagglutinin gene is out of the scope of this manuscript.